# Hypnosis Sedation Reduces the Duration of Different Side Effects of Cancer Treatments in Breast Cancer Patients Receiving Neoadjuvant Chemotherapy

**DOI:** 10.3390/cancers13164147

**Published:** 2021-08-18

**Authors:** Martine Berliere, Nathan Piette, Marion Bernard, Camille Lacroix, Amandine Gerday, Vasiliki Samartzi, Maude Coyette, Fabienne Roelants, Marie-Agnes Docquier, Nassim Touil, Christine Watremez, Philippe Piette, Fran×ois P. Duhoux

**Affiliations:** 1Breast Clinic, King Albert II Cancer Institute, Cliniques Universitaires Saint-Luc, 1200 Woluwe-Saint-Lambert, Belgium; marion.bernard@student.uclouvain.be (M.B.); camille.lacroix@student.uclouvain.be (C.L.); amandine.gerday@uclouvain.be (A.G.); maude.coyette@uclouvain.be (M.C.); francois.duhoux@uclouvain.be (F.P.D.); 2Department of Oncology Cliniques Saint-Pierre, 1340 Ottignies, Belgium; nathan.piette@student.uclouvain.be; 3Department of Gynaecology, Hôpital de Jolimont, 7100 La Louvière, Belgium; vassiliki.samartzi@jolimont.be; 4Department of Anesthesiology, Cliniques Universitaires Saint-Luc, Université Catholique de Louvain, 1340 Ottignies-Louvain-la-Neuve, Belgium; fabienne.roelants@uclouvain.be (F.R.); marie-agnes.docquier@uclouvain.be (M.-A.D.); nassim.touil@uclouvain.be (N.T.); christine.watremez@uclouvain.be (C.W.); 5Medical and Financial Department, Grand Hôpital de Charleroi, 6000 Charleroi, Belgium; Philippe.piette@ghdc.be

**Keywords:** polyneuropathy, postoperative pain, musculoskeletal pain, cancer-related fatigue, neoadjuvant chemotherapy, hypnosis sedation

## Abstract

**Simple Summary:**

Reducing side effects of cancer treatments is a major challenge for clinicians involved in the management of breast cancer patients. Among patients receiving neoadjuvant chemotherapy followed by surgery, radiotherapy and endocrine therapy, prolonged side effects frequently mentioned are: polyneuropathy, musculoskeletal pain, postoperative pain and cancer-related fatigue. Conventional drugs have proven to be ineffective in treating theses effects, except for postoperative pain. This is the reason why we prospectively tested the impact of hypnosis sedation used as anesthetic technique for breast cancer surgery on the different side effects of cancer treatment. Despite the limitations of this small non-randomized cohort, preliminary results are very encouraging.

**Abstract:**

Background: Reducing side effects of cancer treatments is a major challenge for clinicians involved in the management of breast cancer patients. Methods: We analyzed data from 63 patients (32 in the general anesthesia group and 31 in the hypnosis sedation group) who were included in 1 prospective non-randomized trial evaluating hypnosis sedation in breast cancer treatment. The patients were followed every 3 months for 2 years. All patients received neoadjuvant chemotherapy with 4 cycles of epirubicin and cyclophosphamide followed by taxanes. Thereafter, patients underwent surgery while on general anesthesia or while on hypnosis sedation. Radiotherapy was administered according to institutional guidelines. Endocrine therapy was prescribed if tumors expressed hormone receptors. Prevalence, intensity and duration of polyneuropathy, musculoskeletal pain, postoperative pain and cancer-related fatigue were assessed at each medical visit. Results: Symptoms duration was statistically reduced for polyneuropathy (*p* < 0.05), musculoskeletal pain (*p* < 0.05) postoperative pain and cancer-related fatigue (*p* < 0.05) in the hypnosis group. Conclusion: Despite the limitations of this study (lack of randomization and small size) we conclude that hypnosis sedation may exert a role on different side effects of breast cancer treatment in patients receiving neoadjuvant chemotherapy, mainly by reducing their duration.

## 1. Introduction

Breast cancer is the most frequent cancer in women, and the second primary cause of cancer-related deaths. During the last decade, multidisciplinary approach and therapeutic progress have increased overall survival. However, survival benefits have a price. Side effects of anticancer treatments exert deleterious effects on quality of life. As an increasing number of patients are surviving breast cancer, it is of major importance to preserve the best quality of life and thus to reduce therapy-induced side effects [1,2]. In this study, we focused on the early and late long-term symptoms after neoadjuvant chemotherapy. We evaluated the frequency, severity and persistence of the following symptoms:

Polyneuropathy—also called chemotherapy-induced peripheral neuropathy—is a common neurotoxic effect of antineoplastic agents such as taxanes. It refers to pain in the extremities associated with positive and/or negative symptoms like paresthesia and numbness [3,4,5,6,7,8,9].

Musculoskeletal pain is clearly associated with endocrine therapy but also described during and after administration of chemotherapy. Musculoskeletal pain refers to pain originating from or resembling arthralgia and/or myalgia [10].

Postoperative pain needs to be carefully evaluated and treated. It concerns pain located in the surgical area: breast in case of conservative approach, chest wall in case of mastectomy, shoulder and arm. The perioperative period appears as a key time point to prevent the development of chronic pain after the classical period of postoperative pain [2]. Persistent pain following treatment of breast cancer is a common occurrence affecting 25 to 60% of patients and has been linked to decreased quality of life [11,12,13,14,15]. In addition to surgery, neoadjuvant and adjuvant therapies such as systemic treatments and radiation can also lead to persistent pain [16,17,18,19].

Cancer-related fatigue (CRF) is one of the most commonly reported symptoms impacting cancer survivors. CRF is estimated to occur in up to 90% of patients during active treatment and 27 to 82% of patients after treatment. CRF is defined as multidimensional and distressing fatigue related to cancer and/or cancer treatment that interferes with activities of daily living [20,21,22,23]. It can negatively impact multiple facets of a cancer survivor’s life, resulting in decreased quality of life [22].

Advances in pharmacology are certainly required to reduce postoperative pain and its transition to chronicity. In addition, innovative approaches are needed to reduce the incidence of CRF, PNP and MSP, for which a purely pharmacological approach has proven particularly disappointing.

## 2. Materials and Methods

Patients were recruited from 1 study performed in our breast clinic (King Albert II Cancer Institute, Cliniques universitaires Saint-Luc-Université catholique de Louvain) and in the Breast Clinic, Hopital de Jolimont, conducted to evaluate the benefits of hypnosis sedation in breast cancer patients undergoing oncologic surgery. Thereafter, substudies focused on benefits of hypnosis according to different modalities of received treatments such as neoadjuvant chemotherapy. This study was a non-randomized trial approved by our 2 local ethics committees (Comité d’Ethique Hospitalo Facultaire Saint-Luc-UCL and Comité d’Ethique Jolimont-Lobbes; 2016/08JUL/311, 2017/13OCT and 2017/30OCT, B403201629079) and complied with the Declaration of Helsinki. It was registered on clinicaltrials.gov with NCT03330717.

This study was a prospective study registered in Belgium on the clinicaltrials database and registered on clinicaltrials.gov in October 2017 when we decided to open the second center; the first patient of the multicentric study was enrolled in November 2017 and the study was completed at the end of 2019. It included 284 patients. A total of 63 patients received neoadjuvant chemotherapy: 32 in the group of general anesthesia and 31 in the group of hypnosis sedation. Written informed consent was obtained from all patients.

Because of interesting results observed in a first observational study [1], we retained the hypothesis that hypnosis sedation can reduce the duration of side effects, such as polyneuropathy, MSP and CRF linked to chemotherapy. We also focused on the duration of postoperative pain and intensity of pain at different points of measurement. The group of general anesthesia was subdivided in two subgroups: general anesthesia alone (*n* = 25) and general anesthesia preceded by a session of virtual reality (hypnorelaxation-Aqua Program, Oncomfort, *n* = 7) We are aware that this study presents different biases. It could not be randomized because of the distressing context of a recent breast cancer diagnosis. Highly motivated patients who wanted to undergo breast surgery while on hypnosis sedation were not ready to participate in a randomized trial. On the contrary, patients afraid of this technique did not want to take the risk of breast surgery while on hypnosis sedation. Because of the small sample size of the cohort, we only considered 2 arms: general anesthesia (*n* = 32) and hypnosis sedation (*n* = 31).

### 2.1. Population and Data Collection

Eligible patients were patients included in our prospective trial who had neoadjuvant chemotherapy as initial treatment for their breast cancer. A majority were treated with 4 cycles of epirubicin-cyclophosphamide (EC) followed by 12 cycles of weekly paclitaxel, while 10 patients received 4 cycles of EC followed by 4 cycles of docetaxel. A total of 63 patients were evaluated. Twenty patients received trastuzumab during the paclitaxel administration. Dose dense administration of EC was given to 50 patients, with support of G-CSF, and 13 patients received were administered EC every 3 weeks, without support of G-CSF.

In both groups, patients were evaluated before surgery during a preoperative anesthesiology consultation. The modalities and the course of the procedure were extensively described to patients. In this study, no patient requesting hypnosis sedation was refused [1,2].

Clinical data such as medical factors were gathered from medical records reviewed by trained research assistants. When performing measures of the different parameters, they were not aware of the allocated group in which patients were included.

This substudy dedicated to patients receiving neoadjuvant chemotherapy focused on 4 side effects: polyneuropathy, musculoskeletal pain, postoperative pain and cancer-related fatigue. The main objective was to study the duration of the different side effects in the two groups. The secondary objectives were to evaluate the prevalence and the severity of the different side effects. No drop out was observed during the follow-up period. Only one patient in the GA group prematurely stopped her endocrine therapy. We considered this study as a “preliminary prospective study” whose aim was to attest the benefits of hypnosis sedation on different side effects of anticancer treatments.

### 2.2. Description of Hypnosis Sedation Procedure and General Anesthesia

In both groups, patients were evaluated before surgery during a preoperative anesthesiology consultation. In the HYP group, patients received specific explanations about hypnosis sedation. During this session, the modalities and the course of the procedure were described to patients, and physicians confirmed that they were adequate candidates for this kind of analgesia and anesthetic procedure, i.e., that they were able to sign an informed consent form and able to understand the languages spoken in our institution. No patient requesting hypnosis sedation was refused. One hour before surgery, premedication with lorazepam (0.5 mg) was proposed to the patient. At the time of the surgical procedure, all the patients were monitored classically (electrocardiography, noninvasive blood pressure measurement, blood oxygen saturation assessment [SpO_2_], and capnography). Local anesthesia was performed with a combination of levobupivacaine 0.25% and lidocaine 1%. Oxygen was administered to each patient. Once they were comfortably installed on the operating table, the anesthesiologist induced hypnosis as a procedure where indirect suggestions were given on the anesthesiologist’s observation of patient’s behavior, and on her or his judgement of the patient’s needs. The patients were invited to fix a point in front of them while concentrating on their body to achieve total muscle relaxation before finally closing their eyes. Guided by the anesthesiologist, the patients had to focus their attention on a positive recollection. By using a calm and monotonous voice, the anesthesiologist constantly talked to help them relive a dream or experience so that they remained as detached and dissociated as possible from the reality surrounding them. A state of intense well-being and comfort had to be reached and maintained during the whole procedure. The peri-incisional skin is injected with a local anesthetic such as 0.5% lidocaine combined to 0.25% levobupivacaine. A continuous infusion of remifentanil, a µ-opioid agonist, was started at a rate of 0.05 µg/kg/min (a dose about 10 times lower than the one used for general anesthesia) and was modified or stopped as required. If needed, small doses of midazolam were administered, 0.1 mg at a time if an anxiolytic was needed. A preestablished communication system between the anesthesiologist and their patients allowed them to express any discomfort. In such a case, the hypnotic state was strengthened, the surgeon could improve local anesthesia, or the infusion rate of remifentanil could be increased. Once the procedure was completed, the anesthesiologist gave the patients recommendations (posthypnotic suggestions) in order to preserve their comfort in the postoperative period, to have correct healing, to keep the wound dry, and to give the patient the opportunity to reuse hypnosis during their cancer treatment. 

None of the patients in the HYP group included in the current study required a conversion to general anesthesia. In this group, patients thus maintained consciousness during the whole surgical procedure and avoided pharmacological coma. General anesthesia was performed following the usual institutional procedures, based on interventional guidelines.

Premedication with lorazepam was the same in the 2 groups. In an attempt to reduce bias, pre- and postoperative suggestions were given by the anesthesiologists to patients undergoing surgery while on general anesthesia. Local anesthesia was injected before skin incision. General anesthesia was induced by intravenous administration (continuous infusion) of propofol (2–3 mg/kg), with lidocaine hydrochloride (1 mg/kg) ketamine hydrochloride 0.3 mg/kg, and sufentanil of 0.1–0.2 µg/kg and cis-atracurium if necessary. The airway was secured with an endotracheal or supraglottic tube, and the lungs were ventilated with a mixture of oxygen and air (50%-50%); The tidal volume was set at 6–8 mL/kg ideal body weight. Anesthesia was maintained with intravenous administration of propofol (Target controlled infusion devices), and additional intravenous sufentanil citrate (5 µg) was administered during surgery if the heart rate or surgical blood pressure increased by more than 20%. In the days after surgery, pain was controlled following the institution’s protocol: paracetamol 1 g/6 h and naproxen 500 mg/12 h in case of low pain, tramadol 50 mg/6 h in case of mild pain, and piritramide 20 mg/12 h in case of severe pain. Those medicines were given to patients as required.

Patients receiving a virtual session of hypnorelaxation were allocated to the general anesthesia group. The program was administered just before entering the operating room. This program of Oncomfort is named AQUA and corresponds to an immersion in an aquatic environment. The short duration of the session (20 min) is probably responsible for the absence of observation of positive impact on side effects. That is the reason why patients were separated in two groups: general anesthesia and hypnosis sedation.

### 2.3. Polyneuropathy

During administration of chemotherapy, patients were evaluated for polyneuropathy at each course of chemotherapy. They were followed according to the classic procedures used as standard of care in our institution. The questions included the presence and persistence of paresthesias and tingling or pain in fingers or toes [3].

If the answer was yes, severity was assessed and classified between mild, moderate and severe [8]. After surgery, questions were asked to document the presence or absence of polyneuropathy, taken from the EORTC QLQ-CIIPN20 quality of life questionnaire [9].

Questions were repeated at each follow-up visit and the responses were collected by the study’s physician investigators.

### 2.4. Postoperative Pain

In our institution, all patients were evaluated for acute pain during the postoperative period in an attempt to adequately treat acute postoperative pain and avoid chronicization of pain [2]. Evaluations were performed by Numerical Pain Rating Scales and Visual Analog Scales on day 0, day 1 and day 8; thereafter, such as for polyneuropathy, musculoskeletal pain and cancer-related fatigue, monitoring was assessed during oncologic follow-up: every 3 months during the first 2 years after the surgical step.

### 2.5. Musculoskeletal Pain

Musculoskeletal pain is linked to the administration of endocrine therapy and especially to aromatase inhibitors use, with or without Gonadotropin-Releasing Hormone (GnRH) agonists [10].

This type of pain was recorded at each medical visit and also measured by Numerical Pain Rating Scales and Visual Analog Scales. The measurements were carried out by the study’s physician investigators.

### 2.6. Cancer-Related Fatigue

As it is a subjective experience, CRF is measured most efficiently via self-report, and recent observations and analyses suggest that a 10-point Numerical Rating Scale (NRS) for fatigue is the best screening tool. There is an agreement that fatigue intensity is graded as mild with scores of 1–3, moderate between 4 and 6, and severe between 7 and 10 [22,23]. The NRS and intensity score were used in our study and completed by physicians at each medical visit.

### 2.7. Statistical Analysis

Data were analyzed using the R Core Team software, 2021 (R Foundation for Statistical Computing, Vienna, Austria; https://www.r-project.org, accessed on 24 May 2021) [24]. *p* values < 0.05 were considered as statistically significant.

The Mann Whitney *U*-test was used to compare demographic data and tumor characteristics between the two groups and to confirm the homogeneity of the two groups. It was also used to compare pain ratings at different points of measurement. To compare the prevalence, the severity and duration of side effects, a chi square test, a student test and a Welch Two sample *t*-test were used.

## 3. Results

Table 1 presents the patients and tumor characteristics of the 63 patients.

Table 2 summarizes the modalities of treatment.

Table 3 presents prevalence, severity and duration of side effects.

Table 4 presents average pain ratings at the different measurement points.

Figure 1 compares the average duration of the studied side effects observed in the 2 groups according to their time of onset. The time of onset of a side effect is calculated in relation to the date of the surgery (dotted vertical line). The average duration of side effects in the hypnosis sedation group is systematically lower than the average duration observed in the general anesthesia group.

The two groups of patients studied had similar demographic and tumor characteristics. The treatment modalities did not exhibit any significant differences. Full data on the side effects after treatments are available in the Appendix A.

The observed incidence of PNP was 65% (21/32) in the GA group and 58% (18/31) in the HYP group. No statistical difference was noted between the two groups of patients. Among the patients suffering from PNP, 3 patients had severe symptoms in the GA group and 2 in the HYP group. The severity of the symptoms required premature interruption of taxanes administration in 5 patients. Two patients had received 10 courses of paclitaxel, 1 patient had received 9 courses, and 2 patients had received 3 cycles of docetaxel. The duration of PNP was statistically reduced in the HYP group: *p* < 0.05.

The incidence of MSP was 46% (15/32) in the GA group and 29% (8/31) in the HYP group. The incidence and intensity of pain were not statistically different between the two groups but the duration of this symptom was also reduced in the HYP group: *p* < 0.05.

POP was mentioned by all patients on day 1 but severity was lower in the HYP group: 0/31 severe and 0/31 moderate pain. On the contrary, in the GA group, 16 patients described moderate pain and 16 mild pain. On day 8, 31/32 patients of the GA group were still consuming non-steroidal anti-inflammatory drugs while in the HYP group only one patient took the same drugs. Pain intensity was significantly reduced in HYP group on days 0, 1 and 8 (Table 4). Duration of POP was also reduced in the HYP group: *p* < 0.05. 

CRF was cited by all patients at the end of neoadjuvant chemotherapy, but like for the other studied side effects, the duration was statistically reduced in the HYP group: *p* < 0.05.

## 4. Discussion

As soon as women are diagnosed with breast cancer, their quality of life is affected: sadness, anxiety, fatigue and depression are symptoms presented by breast cancer patients that lead to a decreased quality of life [12]. Typically, these negative emotions are added to the difficulties of patients trying to cope with the side effects of their treatment. Polyneuropathy [6,7,8], musculoskeletal pain [10] and postoperative pain [14,15] are directly correlated with the type of treatment administered.

The major limitations of this study are the small number of included patients and the fact that this study is not randomized [1,2]. In our clinical experience, patients who wish to undergo surgery under hypnosis sedation refuse to participate in a randomized study in which they would risk surgery under general anesthesia [1]. Patients communicated their decision to undergo breast surgery while on hypnosis sedation at the beginning of their treatment and no change in anesthesia modality was observed in the study. Patients highly motivated by hypnosis probably have a different psychological profile but all the differences observed in the duration of side effects cannot be attributed exclusively to psychological personality.

The positive points of this study are represented by a very regular follow-up during the administration of neoadjuvant chemotherapy, the perioperative period and radiotherapy. Thereafter, the follow-up period is very carefully monitored and identical for all patients. In general, large prospective randomized trials dedicated to neoadjuvant therapies do not focus on late side effects of cytostatic agents. On the contrary, in this study, late toxic side effects were taken into account.

Neuropathy, also called chemotherapy-induced polyneuropathy, is a disabling side effect of chemotherapy which may greatly affect patients’ quality of life [4,5]. The incidence of neurotoxicity in this study (65% in the GA group and 58% in the HYP group) is comparable to the incidence described in other studies (Ciruleos and MATOX project) [6,7]. The study presented by Ciruleos concerns metastatic breast cancer patients receiving three dose regimens of Nab-paclitaxel [7]. According to the dose escalation, the authors found an incidence of PNP comprised between 50 to 81%. They used a score named total neurotoxicity score to calculate this incidence. As previously mentioned, patients involved in this study were metastatic breast cancer patients and long-term toxicity was not assessed. On the contrary, in the MATOX project, we have data after three years of follow-up [6]. In this trial, 453 patients were evaluated for long-term toxicities of chemotherapy, with 58 patients in the same context as our patients because they received neoadjuvant chemotherapy. In this study, 30% of patients exhibited paresthesia symptoms four weeks after the beginning of chemotherapy and 60% of patients still complained of polyneuropathy at three years of follow-up. Not all included patients received schedules of chemotherapy with taxanes. In the study of Forget, polyneuropathy was measured one and two years after surgery [25]. The prevalence of PNP was respectively 22% and 10% of patients one and two years after surgery. The differences observed in the study of Forget could also be related to the fact that taxane-based chemotherapy was not administered to all patients.

No therapies are approved by the US Food and Drug Administration (FDA) or the European Medicines Agency (EMA) to prevent or treat chemotherapy-induced polyneuropathy [26,27]. Drugs such as gabapentin, duloxetine and vitamins used to decrease the severity and duration of polyneuropathy are associated with limited success. It is thus generally admitted that therapeutic options for CIPN are still limited, and pharmacological treatment focuses on reduction or relief of neuropathic pain [26,27,28]. It is therefore essential to detect chemically-induced polyneuropathy early in order to prevent the development of severe forms and to limit the duration of toxicity over time [29,30]. Currently, there is no causative proven therapy for the prevention of CIPN [27,30]. Pharmacological research needs to intensify its efforts and study new molecules. On the other hand, quality of life issues and rehabilitation concepts for long-term deficits will also be future directions for research. To reduce pain, exercise, acupuncture [31], behaviorally based techniques and hypnosis could be useful [32]. We need to keep in mind that perception of neuropathic pain may be increased by anxiety and depression. Holistic therapeutic management can therefore clearly claim its place. 

Postoperative pain (POP) is a very important parameter which needs to be taken into consideration [2,14,15,16]. The incidence of acute POP was similar in the two groups (GA and HYP), but the duration of postoperative pain was lowered in the HYP group (*p* < 0.05). Pain severity was also significantly reduced in the HYP group (*p* < 0.05), according to the time point of pain measurement This observation confirmed data highlighted in our observational study [1,2]. These data are noted in Table 4. Different pharmacological options have been extensively studied in order to control POP and avoid chronic pain [33,34]. This entity was often described as post mastectomy pain syndrome and more recently as the persistent post-surgery pain syndrome, because patients undergoing lumpectomies were also concerned by this type of pain [11,19]. In the MATOX study, at 3 years, 67% of the treated patients reported persistent post-surgical pain (41% mild pain, 22% moderate pain and 4% strong pain) [6]. In a systematic review published by Wang, the authors mention a prevalence of 27.3% of persistent pain among patients receiving radiotherapy [11]. In this review, different problems were identified: the great variety of treatments received, the variations in follow-up duration and the different definitions of persistent pain contributed to highly variable reports of pain. Many different strategies have been developed to avoid chronic pain after breast surgery: use of different drugs, local anesthesia, interpectoral blocks. More recently, hypnosis exhibited interesting benefits in reducing the prevalence of persistent pain after breast surgery and especially mastectomy [35,36,37].

In a recent study published by Forget (pre-planned sub-study of the Ketorolac in Breast Cancer trial-KBC), the authors reported an incidence of permanent pain at one year comprised between 65 and 75% and incidence of persistent pain between 59 and 63% at two years [25]. The authors mentioned that the description and location of pain changed during the second postoperative year [25].

In our study, immediate postoperative pain was well-controlled with pure analgesic and anti-inflammatory drugs. In the HYP group, the duration of POP was statistically reduced and consumption of anti-inflammatory drugs limited to day one post-surgery, except for one patient suffering from polyarthritis. It is well recognized that shortened duration of POP is a very important parameter to avoid persistent pain after breast surgery. As mentioned in the study of Forget, the location of POP changes after one and two years and it sometimes becomes difficult to characterize pain [25]. Intrication with musculoskeletal pain is not infrequent [25].

Concerning musculoskeletal pain, in the Matox project [6] and in our study, this symptom is clearly linked with the use of endocrine therapy and especially with the use of aromatase inhibitors (taken alone or in association with GNRH agonists) [10]. This type of pain is still present at three years in 40% of the patients included in the Matox project. In our study, the incidence is reduced in the HYP group but the difference is not statistically significant; on the contrary, the duration is shortened in the HYP group and as for other symptoms, reduction of duration is statistically significant. As with chemically-induced polyneuropathy, the success of drug therapies for musculoskeletal pain is limited and generally temporary.

In a recent review, Fabi reported that 40% of patients mentioned fatigue at cancer diagnosis [20]. Eighty to 90% of all patients treated with chemotherapy, surgery, radiotherapy and endocrine therapy reported to suffer from CRF at different levels of intensity. Until now, the etiology of CRF has not yet been elucidated: cytokines have been implicated in the pathophysiology of fatigue, probably by acting at multiple levels including mood, muscle mass and metabolic status [38,39]. In our study, during chemotherapy administration all patients reported fatigue but at different levels. We observed a not statistically significant decrease of incidence and severity of CRF in the HYP group but the duration of CRF is statistically reduced in this group. Different meta-analyses and reviews have concluded that—except for metastatic cancer patients—drugs are totally inefficient and contraindicated because they could be dangerous and toxic [2,21,22,23]. Other strategies—such as behavioral, cognitive therapies and acupuncture need to be developed [40,41,42,43].

A possible explanation of reducing the duration of PNP, POP, MSP and CRF in the HYP group is the fact that we have demonstrated in our prospective study (preliminary results presented in poster session, SABCS, 2018) [44] that hypnosis sedation is able to reduce the duration and the intensity of the inflammatory process linked to the surgical procedure. Inflammation is clearly associated with cancer development and evolution [45,46,47,48]. It represents an interesting target to treat cancer and to decrease the intensity of side effects of anticancer therapies [48,49,50]. Biological measures of the inflammatory reaction were performed using dosage of C-reactive protein (CRP) and neutrophil-to-lymphocyte ratio (NLR) on days 0, 1 and 8 after surgery. Patients who underwent breast surgery while on hypnosis sedation and local anesthesia had statistically lower values of CRP and NLR on day 1. No significant differences were noted on day 0 and day 8 between the two groups of patients (general anesthesia and hypnosis sedation). The positive impact of hypnosis sedation on inflammatory parameters has already been explored by Th. Defechereux et al. [51]. They published a randomized trial evaluating hypnosis sedation in cervical surgery. The authors demonstrated an early lowering of CRP on day 0 (6 hours after the surgical procedure) and of Interleukin-6 (IL-6) on day 1.

Inflammation is implicated in different types of studied pain: polyneuropathy, postoperative pain and musculoskeletal pain, and its role is suspected in the concept of cancer-related fatigue [50]. Thus, reducing the inflammatory process is certainly key to decrease side effects of anticancer treatments. This is the reason why some research protocols have used anti-inflammatory drugs such as non-steroidal anti-inflammatory medications. These drugs are efficient to decrease acute postoperative pain but less efficient to attack long-term toxicities of cancer treatments such as musculoskeletal pain and polyneuropathy. In the study published by Forget, no difference was noted between the group receiving ketorolac and the placebo group, but only one shot of ketorolac was administered [25]. According to the results of our study, hypnosis sedation is able to reduce the inflammatory reaction related to the surgical procedure.

Hypnosis has already been used successfully in metastatic breast cancer to relieve pain. In the adjuvant setting, hypnoanalgesia has been shown to reduce the intensity of the side effects induced by radiotherapy [40] (fatigue, pain, inflammation), by chemotherapy (nausea and vomiting) and by endocrine therapy (hot flashes, musculoskeletal pain) [1,34]. It is therefore not very surprising that hypnosis sedation proves to be able to mitigate polyneuropathy, postoperative pain, musculoskeletal pain and fatigue. It is also necessary to remember that these symptoms are closely intertwined with anxiety, sadness and depression, which do not respond optimally to conventional pharmacological interventions [52,53,54,55,56]. Concerning the addition of virtual reality sessions before general anesthesia, we are convinced that the duration of the sessions needs to be increased and the programs diversified. The small number of patients in this substudy does not allow us to draw any conclusions.

These encouraging results open the door to new research protocols evaluating in a randomized way the effectiveness of hypnosis from the initiation of neoadjuvant treatments.

## 5. Conclusions

The side effects analyzed in this study can have a prolonged duration and profoundly alter the quality of life of patients. It is therefore particularly important to develop a holistic view to manage these side effects and reduce their duration and intensity [56,57]. Unfortunately—with the exception of the management of acute postoperative pain—available drug therapies have proven to be not very effective and not without toxicity. It thus seems to be very important to develop alternative and efficient therapies.

We are aware of the limitations and selection bias of our study but nevertheless, we think that the highly promising obtained results open the door to the design of new prospective trials.

Hypnosis certainly has its place and deserves to be studied in large randomized trials: hypnosis sessions proposed before each course of chemotherapy versus sham procedures. In order to study its mechanisms of action, biomarkers not influenced by chemotherapy administration should be evaluated.

## Figures and Tables

**Figure 1 cancers-13-04147-f001:**
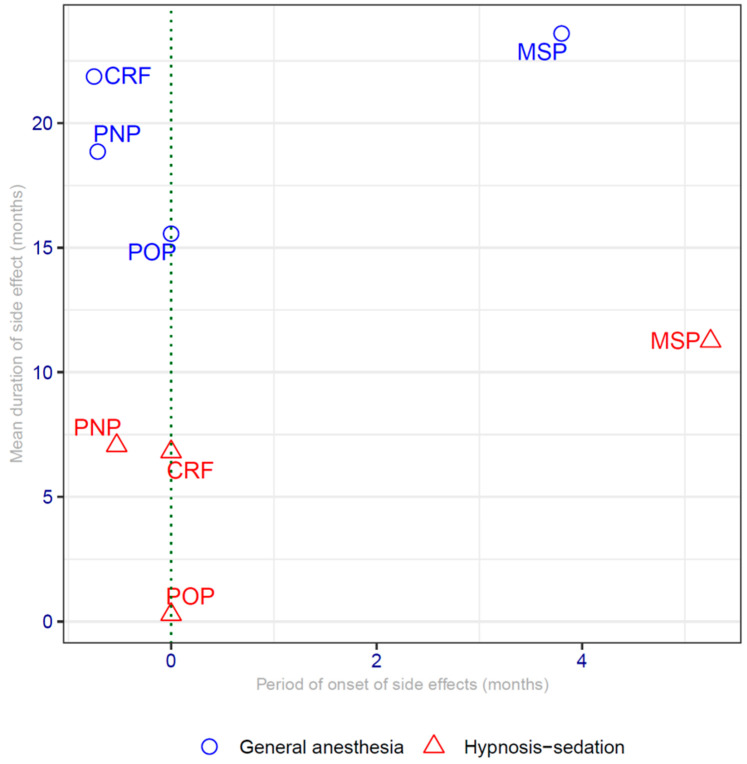
Timing and duration of side effects. CRF = cancer-related fatigue; MSP = musculo-skeletal pain; POP = post-operative pain; PNP= polyneuropathy; dotted vertical line: 0 = time of the surgery.

**Table 1 cancers-13-04147-t001:** Patients and tumor characteristics.

Characteristics	GA Group (*n* = 32)	HYP Group (*n* = 31)
Mean age	53	-
Standard deviation	11.39	-
Menopausal status		
Pre	15	16
Post	17	15
Medical history of		
Polyneuropathy	2	1
Depression	2	2
Polyarthritis	1	1
Histological subtype		
IDC	30	29
Mixed (IDC + ILC)	2	2
Hormone receptors		
HR+	17	15
HR−	15	16
HER2 SISH+	10	10
Triple negative	10	11

GA = general anesthesia; HR = hormone receptor; HYP = hypnosis sedation; IDC = invasive ductal carcinoma; ILC = invasive lobular carcinoma; SISH = silver stain in situ hybridization.

**Table 2 cancers-13-04147-t002:** Treatment modalities.

Characteristics	GA Group (*n* = 32)	HYP Group (*n* = 31)
Chemotherapy regimen		
4 EC/4 Docetaxel	5	5
4 EC dd/12 Paclitaxel	20	20
4 EC/12 Paclitaxel	7	6
Surgery		
Breast conserving	28	29
Mastectomy	4	2
Sentinel lymph node dissection	18	19
Axillary dissection	14	12
Radiotherapy		
Yes	30	30
No	2	1
Traztuzumab alone	8	7
Trastuzumab + pertuzumab	2	3
Endocrine therapy		
Tamoxifen	2	2
Aromatase inhibitors	9	7
Aromatase inhibitors + GnRH agonists	3	3
Tamoxifen + GnRH agonists	3	3

dd = dose-dense; EC = epirubicin–cyclophosphamide; GnRH = Gonadotropin-Releasing Hormone.; GA= general anesthesia; HYP= hypnosis sedation

**Table 3 cancers-13-04147-t003:** Prevalence and difference of duration of side effects studied.

Characteristics	GA Group (*n* = 32)	HYP Group (*n* = 31)
PNP	21 (65%)	18 (58%)
Grade III	3 (9%)	2 (6%)
Difference duration	*p* = 5 × 10^−11^	
95% confidence interval	(9.4–14.2)	
Mean duration (months)	18.9	7.1
MSP	15 (46%)	8 (29%)
Difference duration	*p* = 1 × 10^−8^	
95% confidence interval	(10–14)	
Mean duration (months)	24	11
POP	All patients	All patients
Difference duration	*p* = 2 × 10^−16^	
95% confidence interval	(13–17)	
Mean duration (months)	15.6	0.28
Consumption of NSAIDs >8 days	31/32	1/1
CRF	All patients	All patients
Difference duration	*p* = 2 × 10^−16^	
95% confidence interval	(14–16)	
Mean duration (months)	21.9	6.8

CRF = cancer-related fatigue; GA = general anesthesia; HYP = hypnosis sedation; MSP = musculoskeletal pain; NSAIDs = non-steroidal anti-inflammatory drugs; PNP = polyneuropathy; POP = postoperative pain.

**Table 4 cancers-13-04147-t004:** Average pain ratings at the different measurement points.

Moment	Mean POP HYP	SD HYP	Mean POP GA	SD GA	*p*-Value
D0	0.580	0.501	2.375	0.491	1.58 × 10^−12^
D1	1.096	0.3005	2.531	0.507	3.08 × 10^−12^
D8	0.741	0.444	4	0.254	1.11 × 10^−13^
M3	NA	NA	2.593	0.614	NA
M6	NA	NA	2.187	0.859	NA
M9	NA	NA	2.655	0.720	NA
M12	NA	NA	1.862	0.915	NA
M15	NA	NA	2.428	0.646	NA
M18	NA	NA	2.428	0.646	NA
M21	NA	NA	2.857	0.690	NA
M24	NA	NA	2.857	0.690	NA

D0, 1, 8 = day 0, 1, 8; SD = standard deviation; M= month; NA= not assessed; POP= postoperative pain; HYP= hypnosis sedation; GA= general anesthesia.

## Data Availability

The data reporting the results will be attached to the manuscript in the Appendix A and can be opened with LibreOffice application.

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
