# Peer review of "Hypnosis Sedation Reduces the Duration of Different Side Effects of Cancer Treatments in Breast Cancer Patients Receiving Neoadjuvant Chemotherapy"

_cancers, 2021, doi:10.3390/cancers13164147_

Round 1
Reviewer 1 Report
This manuscript describes a retrospective re-analysis of data gathered in a previous prospective clinical trial. In this study the authors investigate the differences between two patient groups: a group who got general anesthesia for their breast cancer surgery, compared to a group getting hypnosedation for their surgery. Patients in the hypnosedation group experienced side-effects for a much shorter period of time compared to the other group. The findings indicate that hypnosedation is a wonderful tool for preventing the treatment side-effect for getting persistent. Unfortunately, there are many potential sources of bias in this project that are not addressed, chief among them is potential selection bias, and experimenter biases. Nevertheless, the study shows a clear path for future prospective confirmatory investigation of these effects. Also, the topic of the manuscript is highly relevant for the readership of the journal.
- When discussing eligibility, the authors state that “Patients were eligible if they had been included in the previous studiy and had neo-adjuvant chemotherapy as initial treatment for their breast cancer”. What do the authors mean by “the previous study”? This should be clarified. (also “studiy” should be “study”.) Later, they also note: “In these two studies, no patient requesting hypnosedation was refused.” but it is again unclear why are the references previous studies relevant here. How do they relate to this study?
- The authors claim that this study was registered on clinicaltrials.gov (and also in a trial registry in Belgium). However, the study groups, outcomes and aims described in the paper don’t match the ones described in the trial registry. So instead of a prospective registered trial, this seems like a post-hoc re-analysis of a subgroup of the original registered trial. This is further supported by the fact that in the methods section the authors are referring to previous studies, and refer to this project as a “substudy”. This should be clarified, explaining the discrepancy between the registry entry and the study described in this manuscript.
- Also, the trial registry indicates that the target sample size for the study was 750, but data collection was stopped at reaching 284. The rationale for the original sample size target and the premature stopping before reaching the target sample size should be explained.
- It should be clearly stated whether outcome assessors (for example the physician investigators) were blind to group allocation.
- In the trial registry the authors state that there will be three study arms. Results of the third study arm should also be reported in the paper.
- It should be clarified, who were offered to get the hypnosis sedation option. What was the eligibility criteria for that, how many people declined this option out of the ones who were offered the option. Does the control group only consist of people who were offered the hypnosis sedation option, or those as well who were not eligible to get hypnosis sedation? This could be a potential source of selection bias.
- Dropout or attrition is not mentioned in the manuscript. This topic should be addressed in the paper, since patients were followed for more than a year after the study. How many participants dropped out of the study, and how were their data handled in the analysis?
- The description of the statistical methods used could be clearer. The authors refer to the “Wilcoxon–Mann–Whitney regression model”. There is a “Wilcoxon–Mann–Whitney test” (also called the Mann-Whitney U test) that is a non-parametric alternative for the t-test, but I have not previously heard of this test being referred to as a regression model. this could be clarified. Also, I don’t know what does “chi square x2” mean. I think the authors refer to the chi-squared test.
- Some readers might not be familiar with the p-values being presented in mathematical notations such as “p = 5e-11”. It would be better to simply state that p < 0.05 (or p< 0.001), since the p-value should not be interpreted beyond whether it reached the threshold of significance or not.
- Figure 1 should be referenced in the main text, and it should have a detailed explanation about what can be seen on the figure, because it is not a standard graph.
- De-identified data used in the statistical analysis should be shared, and the analysis code used to produce the results should also be shared to support analytical reproducibility.
- Effects of hypnosis on the duration of side effects are nothing short of extraordinary. I have seen such effects claimed in anecdotal reports and case studies, but not in a prospective clinical trial. For example, for patients in the regular anesthesia group POP lasted for more than a year on average, while it was only 1 week on average for patients who received hypno-sedation. This difference is so great that it is almost unbelievable for me. Based on previous research I am convinced that hypnoanalgesia works and that hypnosis is effective at mitigating postoperative pain, but this result was unexpected. Could the authors provide more detailed data about the findings here? For example, it would be great to see the average pain ratings at the different measurement points of the project. This could reveal some more nuance to this finding. (The other results are also pretty large, but non so extreme as the result about POP).
- The details of the hypnosis-sedation procedure should be shared in detail so that the study can be exactly replicated. If this is a long text, please, make it available as an online supplement.
- The details of the regular anesthesia procedure should be shared as well.
- The authors note that “The prevalence of PNP was respectively 22% and 10% of patients one and two years after surgery.” by Forget et al 2020. Yet, in this study, the authors found that the average PNP duration in the general anesthesia group was more than 1.5 years! This seems like a contradiction. The patients in the general anesthesia group seem to suffer from these post treatment side effects for longer than the average patient. This could be discussed in more detail.
- The authors note when talking about persistent postoperative pain that “In our study, the incidence is reduced in the HYP group but the difference is not statistically significant” I am not sure how this is possible, if the average duration of POP was only a few days compared to more than a year in the general anesthesia group, why is the difference in the incidence of persistent POP not statistically significant? I did not find the relevant data in table 3.
- There are multiple typos in the text and the tables that should be corrected.
Author Response
Response to reviewer I
We thank the reviewer for his comments, suggestions and requirements.
We are aware that our project contains different sources of bias which were mentioned in the conclusion. We have added sources of bias in the material and methods section (page 2) and highlighted them in the conclusion.
The first bias is the fact that this study is a non-randomized study. In the distressing context of a recent breast cancer diagnosis, highly motivated patients, ready to undergo oncological surgery while on hypnosis sedation were not ready to accept a randomized study with the possibility of surgery while on general anesthesia.
On the contrary, patients afraid of hypnosis sedation didn’t accept the risk of this technique and required general anesthesia.
Eligibility: Eligible patients were patients included in this prospective trial who had neoadjuvant chemotherapy as initial treatment (correction has been performed in the text).
The term previous study was incorrect and was suppressed.
Registration: I have sent an email to the assistant editor to explain the “two steps“ registration of our study.
At the end of 2016, we initiated a monocentric study which we registered in the Belgian registry of clinical trials after agreement of our ethics committee. (all the required documents have been sent to the assistant editor). After the first interim analysis, we decided to open the study to a second center (Hôpital de Jolimont) to decrease selection bias. A new agreement was obtained from our ethics committee in July 2017 and the agreement of the second ethics committee was obtained in October 2017 (these documents were sent to the Assistant Editor).
The registration was performed just afterwards and patients included in this substudy were all included after registration on clinicaltrialsgov..
We had originally planned to include 450 patients (750 is a typing error) and to extend the inclusions to 2020, but unfortunately, we had to change our plan due to the covid-19 pandemic. Hypnosis sedation was not possible during the entire first wave and during part of the second wave.
This study actually has 3 arms:
-
1: general anesthesia (GA group).
-
2: general anesthesia preceded by hynorelaxation session (virtual reality: Aqua program by Oncomfort) (GAVR group).
-
3: hypnosis sedation (HYP) group.
Local anesthesia was applied to all patients according to the same modalities.
The two groups of general anesthesia were merged and no difference was observed concerning duration and incidence of side effects in the two subgroups of patients undergoing surgery while on general anesthesia alone or while on general anesthesia plus virtual session of hypnorelaxation.
25 patients received neoadjuvant chemotherapy in the group of general anesthesia alone and 7 in the group of general anesthesia + session of virtual reality .
In this substudy, we had no drop out during the follow-up period. Only one patient stopped prematurely her endocrine therapy in the GA group. This information has been added in the “material and methods” section.
The exclusion criteria for a surgical procedure under hypnosis sedation are psychiatric disorders and severe deafness. In general, patients are referred to our breast clinic because they want to undergo breast surgery while on hypnosis sedation. In this subtudy, no patient was refused and all the procedures performed while on hypnosis sedation were successful.
The description of the statistical analysis has been clarified and corrected. (Mann-Whitney U test and chisquare test).
Concerning p-values, as you suggested, p-values in the text were noted as <0.05 and the mathematical presentation has only been left in Table 3. an Tale 4 (table added)
Detailed explanations have been added to Figure 1
.
The deidentified data will be sent as supplementary material.
The average pain rating at the different measurement points has been noted in a supplementary table (Table 4).The incidence of acute POP is similar in the two groups but severity is decreased inHYP group and incidence of Pesistent POP is reduced .
The physicians responsible for the different measures (PNP, POP, CRF and MSP) were not aware of the allocated group in which patients were included (this information has been added in the text).
The details of the hypnosis sedation procedure and the details of the regular anesthesia procedure have been added in the material and methods section.
The discrepancy between the results performed by Forget and our study could be explained by the fact that in the study of Forget not all patients received taxane-based chemotherapy. That remark has been added in the discussion section.
The incidence of POP was not statistically different in the two groups but the incidence of persistent pain after surgery was reduced in the HYP group, with a p-value <0.05; this remark was also added in the discussion.
Typos were corrected.
Reviewer 2 Report
- The authors need to describe more precisely the aim of this study and what are hypothesis concerning the influence of the anesthetic technique and post-surgical outcomes (early and late long term symptoms.
- The reader of this articles needs more details concerning the specific anesthetic technique "hypnosedation", what is different between general anesthesia and this specific technique.
- Use the same term to describe the hypnosis - sedation à hypnosedation
- Concerning explanations that hypnosedation used during surgery (avoiding pharmacological coma) and that this may influence inflammatory reactions linked to an anesthetic technique need references.I suggest to include the prospective randomized study of Defechereux et al. (Ann Chir 2000).
- In the discussion section, the authors should better discuss their findings and hypothesis, they can help the reader to better understand the pertinence and potential importance of this study.
Author Response
Response to reviewer 2
We thank the reviewer for the remarks, constructive comments and suggestions.
The aim of the study and the hypothesis have been added in the text.
The term “hypnosis sedation” is now used consistently throughout the manuscript.
The 2 procedures (hypnosis sedation and general anesthesia) have been described in the section material and methods.
In the discussion, the impact of hypnosis sedation on inflammatory parameters has been more extensively discussed. The reference of Defechereux, suggested by the reviewer, has been added.
We have tried to further discuss our findings and hypotheses and to improve our conclusion.
Round 2
Reviewer 2 Report
line 414
an early lowering of CRP on day 0 (6 hours after the surgical procedure) and of Interleukin-6 on day 1
line 416
pains